# The Role of Ferroptosis and Cuproptosis in Curcumin against Hepatocellular Carcinoma

**DOI:** 10.3390/molecules28041623

**Published:** 2023-02-08

**Authors:** Zhili Liu, Huihan Ma, Zelin Lai

**Affiliations:** 1Neurosurgery Center, The National Key Clinical Specialty, The Engineering Technology Research Center of Education Ministry of China on Diagnosis and Treatment of Cerebrovascular Disease, Guangdong Provincial Key Laboratory on Brain Function Repair and Regeneration, The Neurosurgery Institute of Guangdong Province, Guangdong-Hong Kong-Macao Greater Bay Area Center for Brain Science and Brain-Inspired Intelligence, Zhujiang Hospital, Southern Medical University, Guangzhou 510282, China; 2Department of Obstetrics and Gynaecology, The Chinese University of Hong Kong, Hong Kong, China; 3Department of Neurology, The Second Affiliated Hospital of Guangzhou University of Chinese Medicine, Guangzhou 510120, China; 4Department of Neurology, Guangdong Provincial Hospital of Chinese Medicine, Guangzhou 510120, China

**Keywords:** curcumin, hepatocellular carcinoma, ferroptosis, cuproptosis

## Abstract

Background: Among cancer-related deaths, hepatocellular carcinoma (HCC) ranks fourth, and traditional Chinese medicine (TCM) treatment is an important complementary alternative therapy for HCC. Curcumin is a natural ingredient extracted from *Curcuma longa* with anti-HCC activity, while the therapeutic mechanisms of curcumin remain unclear, especially on ferroptosis and cuproptosis. Methods: Differentially expressed genes (DEGs) of curcumin treatment in PLC, KMCH, and Huh7 cells were identified, respectively. The common genes among them were then obtained to perform functional enrichment analysis and prognostic analysis. Moreover, weighted gene co-expression network analysis (WGCNA) was carried out for the construction of the co-expression network. The ferroptosis potential index (FPI) and the cuproptosis potential index (CPI) were subsequently used to quantitatively analyze the levels of ferroptosis and cuproptosis. Finally, single-cell transcriptome analysis of liver cancer was conducted. Results: We first identified 702, 515, and 721 DEGs from curcumin-treated PLC, KMCH, and Huh7 cells, respectively. Among them, *HMOX1*, *CYP1A1*, *HMGCS2*, *LCN2*, and *MTTP* may play an essential role in metal ion homeostasis. By WGCNA, grey60 co-expression module was associated with curcumin treatment and involved in the regulation of ion homeostasis. Furthermore, FPI and CPI assessment showed that curcumin had cell-specific effects on ferroptosis and cuproptosis in different HCC cells. In addition, there are also significant differences in ferroptosis and cuproptosis levels among 16 HCC cell subtypes according to single-cell transcriptome data analysis. Conclusions: We developed CPI and combined it with FPI to quantitatively analyze curcumin-treated HCC cells. It was found that ferroptosis and cuproptosis, two known metal ion-mediated forms of programmed cell death, may have a vital effect in treating HCC with curcumin, and there are significant differences in various liver cancer cell types and curcumin treatment which should be considered in the clinical application of curcumin.

## 1. Introduction

Hepatocellular carcinoma (HCC) ranks fourth among cancer-related deaths worldwide, accounting about 90% of all primary liver cancers [1,2]. As an aggressive and malignant tumor, HCC incidence has been rising in recent decades [3,4]. Most confirmed HCC patients are in advanced stage, although radical surgery is the effective therapy for HCC, the effectiveness of which has decreased significantly [5,6]. Besides surgical intervention, drug treatment for HCC is another effective choice. [7]. Current first-line therapy drug against advanced HCC is sorafenib (a multi-kinase inhibitor) [8], but its curative effect is not satisfactory [9,10]. Traditional Chinese medicine (TCM), a complementary alternative therapy, shows significant benefits on prolonging median survival time and improving quality of life among patients with HCC [11].

Curcumin is well-known to be a polyphenol obtained from the plant *Curcuma longa* and has long been used to treat a wide spectrum of chronic diseases, including cancer [12]. Multiple mechanisms have been identified by numerous studies to reveal the anti-liver cancer effects of curcumin. For instance, curcumin could inhibit the proliferation of HCC cells via regulating non-coding RNA [13] and mitochondrial apoptosis [14]. Curcumin also has an impact on the tumor microenvironment [15]. Multiple molecular signaling pathways, including the Wnt/β-catenin [13], PI3K/AKT/GSK-3β [14], and connective tissue growth factor (CTGF) [15] pathways, have been shown to be involved in curcumin-mediated liver protection.

Ferroptosis, identified and named a decade ago, is a unique type of programmed cell death primarily driven by iron-dependent lipid peroxidation [16,17]. Growing evidence has revealed that ferroptosis plays a crucial part in treatment with curcumin against various cancers, such as non-small-cell lung cancer (NSCLC) [18], clear cell renal cell carcinoma (ccRCC) [19], and breast cancer [20,21]. Despite these findings, however, the precise action of curcumin-modulated ferroptosis in HCC remains unclear. As another metal ion-dependent cell death, cuproptosis (a distinct form of copper-induced cell death) was recently reported by Tsvetkov et al. [22]. Some research has suggested an underlying connection between HCC and cuproptosis [23,24]. Therefore, further study on the effect of ferroptosis and cuproptosis in treating HCC with curcumin is necessary.

This study was performed to first explore the transcriptomic change in various HCC cells after curcumin treatment and then explore the involved biological processes, focusing on the underlying role of ferroptosis and cuproptosis in curcumin against HCC. Finally, we investigated the differences in ferroptosis and cuproptosis levels among cell subtypes of HCC.

## 2. Results

### 2.1. Analysis of Differentially Expressed Genes (DEGs) in HCC Cells Treated with Curcumin

Based on the thresholds of |log2 fold change| > 1 and adjusted *p*-value < 0.05, we first identified 702 (421 up-regulated and 281 down-regulated), 515 (263 up-regulated and 252 down-regulated), 721 (395 up-regulated and 326 down-regulated) DEGs from curcumin-treated PLC, KMCH and Huh7 cells, respectively (Appendix A). A volcano plot (Figure 1A) and three heatmaps (Figure 1B–D) were established to exhibit the distribution of DEGs in different liver cancer cell lines with curcumin treatment. Furthermore, we found that 35 genes were identical among the DEGs of the three cell lines (Figure 1E). Among the 35 genes, *CYP1A1*, *HMGCS2*, *HMOX1*, *LCN2*, and *MTTP* could be enriched in iron and copper ion related biological processes (Figure 1F), implicating the curcumin-mediated regulation of metal ion homeostasis through these genes.

In addition, we also analyzed the potential role of 35 DEGs in HCC prognosis. Hence, 1109 genes (*p* < 0.05) associated with HCC prognosis were identified from The Cancer Genome Atlas Program (TCGA) in total (Appendix A). Based on the intersection of 1109 HCC prognosis related genes and 35 DEGs, *APOA1* and *PLAU* were common genes (Figure 2A). Figure 2B shows that high expression of *APOA1* and low expression of *PLAU* were related to the good prognosis of HCC. These results implied that curcumin might regulate other pathways than ferroptosis and cuproptosis to affect prognosis via *APOA1* and *PLAU*.

### 2.2. Weighted Gene Co-Expression Network Analysis (WGCNA) of HCC Cells after Curcumin Treatment

WGCNA was performed in order to build the co-expression network. The power of β = 21 (scale-free R^2^ = 0.85) was selected as the soft-thresholding parameter in order to ensure a scale-free network (Figure 3A). Hence, 20 modules in total were constructed through hierarchical clustering, and the number of genes in each module ranged from 40 to 5429 (Figure 3B). According to the eigengene values (the first principal component of the module gene expression level) of modules, we associated the co-expression modules with traits (Figure 3C). MEgreenyellow, MEmagenta, MEgrey60, MEmidnightblue, and MEsalmon were significantly correlated with curcumin treatment (Figure 3C). Figure 3D shows the correlation among the five modules, and we performed module functional annotation using HumanBase for the five modules. Interestingly, the function of the grey60 module was annotated as cellular chemical homeostasis, regulation of small molecule metabolic process and cellular ion homeostasis (Figure 3D), reflecting the modulation of curcumin for ion homeostasis as well.

### 2.3. Curcumin Promotes Ferroptosis of Part of HCC Cells

Given that iron ion related biological processes was involved in treatment with curcumin against HCC, we further explored the effect of curcumin on ferroptosis of HCC and calculated the Ferroptosis Potential Index (FPI) to assess ferroptosis level of liver cancer cells with or without curcumin treatment. FPI was significantly increased in PLC cells after curcumin treatment (Figure 4A), while remained unchanged in KMCH and Huh7 cells (Figure 4C,E). Heatmaps displayed the expression of genes used to calculate FPI in different cell lines (Figure 4B,D,F). These results suggested that curcumin exerted different effects on ferroptosis due to HCC cellular heterogeneity.

### 2.4. Cuproptosis Potential Index (CPI) Reflects the Cuproptosis Levels of HCC Cells

Except FPI, we developed an index (CPI) to quantitatively assess the level of cuproptosis. In KMCH cells, we found a significant reduction in the CPI of the curcumin treatment group (Figure 5B). Nevertheless, there was no significant change of CPI in curcumin-treated PLC and Huh7 cells (Figure 5A,E). Heatmaps showed the expression of genes used to calculate CPI in different cell lines (Figure 5B,D,F). These results demonstrated that the role of curcumin on cuproptosis was cell-specific as well.

### 2.5. Single-Cell RNA Analysis of HCC Cells on Ferroptosis and Cuproptosis

Considering the cell specificity of ferroptosis and cuproptosis in HCC, we performed single-cell transcriptome analysis. Based on the GSE151530 dataset, we performed a UMAP dimensionality reduction analysis on 56,721 single cells from 46 HCC patients (Figure 6A). After extracting the malignant cells and performing subcluster analysis, we were able to identify 16 subclusters (Figure 6B). By calculating the marker genes of these 16 subclusters, we drew a heatmap and found that the top three marker genes were able to separate the subclusters significantly, indicating that the subclusters were reliable (Figure 6C, Appendix A). We calculated FPI and CPI for each subcluster and observed that the levels of ferroptosis and cuproptosis differed significantly among the different subclusters (Figure 6D,E). These results implied that different types of liver cancer cells had specificity in ferroptosis and cuproptosis levels, which needs to be considered in curcumin treatment.

## 3. Discussion

There is a growing body of convincing evidence that natural compounds could be seen as promising drug candidates for a variety of diseases including cancer. Curcumin, an active component isolated from *Curcuma longa*, exerted anti-cancer activities orchestrated through key signaling pathways related to cancer [12]. Although there were advances in curcumin-mediated anti-liver cancer effects via different mechanisms [13,14,15], challenges remain ahead regarding the understanding of molecular mechanisms that would be pharmacologically necessary to explain the therapeutic potential of curcumin against HCC.

In this study, we first found that curcumin induced changes in the expression of *CYP1A1*, *HMGCS2*, *HMOX1*, *LCN2*, and *MTTP*, which are enriched in metal ion homeostasis, in all three liver cancer cell lines (Figure 1F). In addition, we identified four co-expression modules (greenyellow, magenta, grey60 and midnightblue) associated with curcumin treatment but not with cell type by WGCNA analysis (Figure 3C). Among them, the grey60 module was involved in the regulation of cellular ion homeostasis in the liver. These analyses suggested that ion homeostasis might play a crucial part in treating HCC with curcumin. Ferroptosis and cuproptosis are two forms of metal ion-mediated programmed cell death. It is worth noting that curcumin could affect the ferroptosis pathway in a variety of cancers. For example, in NSCLC, curcumin could induce ferroptosis via autophagic activation to improve the therapeutic efficacy [18]. In ccRCC, curcumin promoted ferroptosis to reverse sunitinib resistance [19]. In breast cancer, curcumin suppressed tumorigenesis via enhancing SLC1A5-mediated ferroptosis [21]. Although there was no direct evidence that curcumin affected cancer development through cuproptosis, a study has shown that cuproptosis-related genes were closely associated with the prognosis and drug sensitivity of HCC [25]. On this basis, these results implied that ferroptosis and cuproptosis may be important pathways in its anti-HCC effect of curcumin.

As a stress-induced enzyme, heme oxygenase-1 (HMOX1) catalyzes the degradation of heme to carbon monoxide, iron, and biliverdin, a potential therapeutic target for many diseases [26]. Indeed, curcumin could activate ferroptosis in breast cancer cells, and *HMOX1* could promote curcumin-induced ferroptosis [20]. EF24, a synthetic analogue of curcumin, induced ferroptosis through up-regulating *HMOX1* in osteosarcoma cells [27]. In our study, the expression level of *HMOX1* gene was significantly raised in all three curcumin-treated liver cancer cell lines (Appendix A), and *HMOX1* was enriched in these biological processes, such as cellular iron ion homeostasis, cellular transition metal ion homeostasis, cellular response to metal ion, iron ion homeostasis, response to iron ion, response to metal ion and transition metal ion homeostasis (Figure 1F), indicating the possibility that curcumin regulated ferroptosis via *HMOX1*.

Microsomal triglyceride transfer protein (MTTP) is highly expressed in adipose tissue and regulates lipid metabolism by promoting the transport of triglycerides between membrane vesicles. A recent study revealed that MTTP protein could inhibit ferroptosis and enhance chemoresistance in colorectal cancer [28]. Moreover, the overexpression of MTTP could reduce hepatic steatosis, inflammation, and fibrosis, implying that it may play an important role in the occurrence and development of HCC [29]. Lipocalin 2 (LCN2), a protein with antioxidant capacity, is upregulated under various cellular stress conditions, particularly in cancer. Valashedi et al. claimed that CRISPR/Cas9-mediated knockout of *LCN2* significantly promoted erastin-mediated ferroptosis and enhanced cisplatin vulnerability in MDA-MB-231 (a kind of human breast cancer cell line) [30]. Yao et al. found that the LIFR/NF-κB/LCN2 axis could regulate the liver tumorigenesis and vulnerability to ferroptosis [31]. In addition, a recent study showed that there was a significant positive correlation between LCN2 protein and copper in serum [32]. Interestingly, several studies have shown that curcumin could obviously inhibit the expression of LCN2 (at both the protein and mRNA levels) in vivo and in vitro [33,34]. These studies suggest that curcumin may regulate ferroptosis and cuproptosis via *LCN2* gene.

HMGCS2, or 3-hydroxy-3-methylglutaryl-coenzyme A synthase 2, is known as a key ketogenic enzyme, mediating ketone production and regulating the proliferation and metastasis of HCC [35]. Recently, Duan et al. claimed that activated PPARa/HMGCS2/NRF2/ARE pathway might induce ferroptosis [36]. In addition, HMGCS2 (at both the protein and mRNA levels) may be involved in ceruloplasmin-mediated copper metabolism [37], implying that *HMGCS2* may play a regulatory role in the process of ferroptosis and cuproptosis. Cytochrome P450 family 1 subfamily A member 1 (CYP1A1) is one of the main P450 enzymes and is highly expressed in the liver and urinary bladder. A recent study reported that kynurenine could provoke the activation of aryl hydrocarbon receptor (AhR), the expression of CYP1A1, and the accumulation of reactive oxygen species (ROS), and ultimately lead to cell ferroptosis [38]. Furthermore, copper ion could significantly induce the expression of *CYP1A1* mRNA [39], suggesting that *CYP1A1* may be involved in cuproptosis.

To further explore the changes in the levels of ferroptosis and cuproptosis after curcumin treatment of hepatoma cells, we calculated FPI and developed CPI (Figure 4 and Figure 5). In addition, we found that some characterized events of ferroptosis were evoked in PLC cells (Appendix A), and several signature pathways of cuproptosis were inhibited in KMCH cells (Appendix A). These results further support the reliability of the FPI and CPI. The results demonstrated that curcumin could significantly promote the ferroptosis level of PLC cells and inhibit the cuproptosis level of KMCH cells, suggesting that curcumin was likely to affect HCC through ferroptosis and cuproptosis pathways. However, ferroptosis had no significant changes in curcumin-treated KMCH and Huh7 cells, and cuproptosis did not change significantly in curcumin-treated PLC and Huh7 cells, reflecting that ferroptosis and cuproptosis in different liver cancer cell types are differentially sensitive to curcumin treatment.

To confirm the cell-type specificity of ferroptosis and cuproptosis in HCC, we analyzed the data of single-cell transcriptome sequencing. Significant differences existed in levels of ferroptosis and cuproptosis among different cell subtypes of HCC (Figure 6D,E). A previous study also showed that HCC cell lines exposed to curcumin reacted differently, which were classified as sensitive or resistant [40]. The dual effects of curcumin on various HCC cell lines may be, at least in part, due to different ferroptosis and cuproptosis levels. Therefore, the specificity of ferroptosis and cuproptosis should be considered in the clinical application of curcumin.

At the phenotypic level, the article (public data source in our study) indicated that curcumin could significantly increase hepatoma cell death (PLC: 18% cell death, KMCH: 31% cell death, Huh7: 41% cell death) in vitro and inhibit the activity of cancer stem cells in vitro and in vivo [40]. These results suggest that curcumin could cause good overall killing effects on these hepatoma cells. In the three curcumin-treated liver cancer cell lines, 702, 515 and 721 DEGs were generated, respectively (Appendix A). However, only 35 DEGs are shared among them (Figure 1E). Moreover, Figure 3C indicated that there are a large number of different gene co-expression networks in different hepatoma cell types, implying significant differences in the response of different cell types to curcumin. In addition, single-cell RNA analysis showed that ferroptosis and cuproptosis were significantly different among multiple liver cancer subtypes (Figure 6). Therefore, on the one hand, curcumin has a different ability to affect ferroptosis and cuproptosis in different liver cancer subtypes, and on the other hand, the anti-cancer effect of curcumin is exerted through multiple pathways.

In our study, we investigated the mechanism by which curcumin affected HCC via ferroptosis and cuproptosis. In addition to the metal ion-mediated programmed cell death, curcumin could affect HCC through other pathways. Curcumin effectively inhibits oncogenic NF-κB signaling and restrains stemness features in liver cancer [40]. Zhou et al. found that curcumin could restrain cell proliferation, migration, and invasion via regulating the miR-21-5p/SOX6 axis in HCC [41]. A recent study reported that curcumin could provoke mitochondrial apoptosis in HCC by BCLAF1-mediated PI3K/AKT/GSK-3β signaling pathway [14]. In addition, the gut microbiota is also involved in the treatment of HCC with curcumin. Jin et al. found that the bioavailability of curcumin could be improved with the help of gut microbiota in HCC [42]. Wu et al. reported that curcumin could inhibit the growth of HCC via regulating gut microbiota-mediated zinc homeostasis [43]. These results suggest that gut microbiota may play an essential role in the process of curcumin affecting ferroptosis and cuproptosis.

Additionally, we found that curcumin may regulate the prognosis of HCC through *APOA1* and *PLAU* (Figure 2). Quantitative proteomic analysis showed that the expression of APOA1 protein was significantly down-regulated by 22% in HCC with hepatitis C virus (HCV) [44]. Guo et al. reported that the expression level of *APOA1* mRNA in the serum of HCC patients with liver cancer was significantly lower than that of healthy people, and high expression of *APOA1* mRNA obviously correlated with better overall survival in the Chinese cohort [45]. These independent studies support the credibility of our results. Although the correlation of *APOA1* with HCC has been confirmed by several studies, the mechanisms of *APOA1* downregulation in liver cancer remain incompletely clear. A study showed that hepatitis B virus (HBV) could induce the DNA hypermethylation of CpG island, leading to epigenetic silencing of *APOA1* gene expression, and then contribute to the pathogenesis of chronic hepatitis B (CHB) [46]. Since CHB is a major risk for liver cancer, it may be one of the mechanisms of *APOA1* downregulation in liver cancer. It is worth noting that the expression of APOA1 protein was seven-fold higher in African American HCC patients compared to Caucasian American HCC patients [47], suggesting that the changes of *APOA1* expression are not only cell type specific, but also race specific. In short, *APOA1* may be a potential therapeutic target for HCC in spite of the fact that further evidence is needed. Interestingly, a study reported that curcumin could increase the expression of *APOA1* in the sorafenib-treated mice and enhance the antitumor effects of sorafenib through regulating the metabolism and tumor microenvironment [48]. In liver cancer, there are not many studies related to the *PLAU* gene. It is worth noting that a study declared that the high expression of serum PLAU protein could be used to predict poor prognosis in HCC after resection [49]. Similarly, our results also showed that a high expression of *PLAU* was related to a poor prognosis in HCC. A study claimed that *PLAU* was associated with chronic inflammation in the liver [50]. Chronic inflammation is an important risk of HCC, so curcumin may improve the prognosis of HCC by regulating the inflammatory response through *PLAU*. These mechanisms are worth further investigation.

## 4. Materials and Methods

### 4.1. Identification of DEGs

GEO database (https://www.ncbi.nlm.nih.gov/geo/, accessed on 24 August 2022) provided DEGs of curcumin (Series: GSE59713, Samples: GSM1444129, GSM1444130, GSM1444131, GSM1444132, GSM1444133, GSM1444134, GSM1444141, GSM1444142, GSM1444143, GSM1444144, GSM1444145, GSM1444146, GSM1444147, GSM1444148, GSM1444149, GSM1444150, GSM1444151, GSM1444152,). In GSE59713, total RNA was isolated from KMCH, Huh7 and PLC cells treated with or without 25 μM curcumin for 72 h in vitro, and three liver cancer cell lines were significantly inhibited by curcumin [40]. Differential analysis was performed by limma R package (v.3.52.2) [51], and the thresholds of DEGs were |log2 fold change| > 1 and adjusted *p*-value < 0.05.

### 4.2. Functional Enrichment Analysis

The biological process of Gene Ontology (GO) analysis was performed via clusterProfiler v.4.4.4 (a kind of R package) [52], and module functional annotation was performed using HumanBase (a data-driven, tissue-specific, and network-guided biological function annotation tool) [53].

### 4.3. Prognostic Analysis

The Cancer Genome Atlas Liver Hepatocellular Carcinoma (TCGA-LIHC) data enabled us to identify 1109 genes (*p* < 0.05) associated with the prognosis of HCC using the Kaplan–Meier (K-M) survival curves. The survival curves of *APOA1* and *PLAU* were visualized using survminer R package, version 0.4.9.

### 4.4. WGCNA

WGCNA is a system biology algorithm commonly used to examine the correlation between gene sets and clinical characteristics by constructing free-scale gene co-expression networks. The expression profile of mRNA was applied to build a gene co-expression network via the package WGCNA implemented in R software, and the version is 1.71 [54]. The WGCNA::blockwiseModules() function was used with the following settings: soft threshold power = 21, TOMType = “unsigned”, minModuleSize = 30, deepSplit = 4 and mergeCutHeight = 0.25.

### 4.5. The FPI Calculating

According to the previous work by Zekun Liu et al., we calculated the FPI [55]. The FPI was used to quantitatively reflect ferroptosis level via transcriptome data.

### 4.6. The CPI Constructing

We constructed the CPI by the expression for essential genes of cuproptosis, including positive components (*PDX1*, *LIAS*, *LIPT1*, *DLD*, *DLAT*, *PDHA1*, *PDHB*) and negative components (*MTF1*, *GLS*, *CDKN2A*) [22]. Through single sample gene set enrichment analysis (ssGSEA), the enrichment score (ES) was calculated, including positively or negatively regulated cuproptosis. Then, normalized differences between the ES of the positive components minus negative components are referred to as the CPI.

### 4.7. Single-Cell RNA Analysis

We performed single-cell RNA analysis of liver cancer based on GEO dataset GSE151530 [56]. GSE151530 is an atlas of liver cancer, and a total of 46 tumor samples from HCC and intrahepatic cholangiocarcinoma patients were profiled. Seurat R package (v.4.1.1) was used for data analysis [57].

## 5. Conclusions

The present study demonstrated that the regulation of ion homeostasis, especially on ferroptosis and cuproptosis, may play a crucial part in the treatment of HCC with curcumin. We quantitatively analyzed different HCC cell lines treated with curcumin and HCC cell subtypes by FPI and CPI. We found that the levels of ferroptosis and cuproptosis are cell specific in various liver cancer cell types and curcumin treatments. Our study is helpful to explore the potential mechanisms of natural compounds and provide preliminary data to assist their clinical application.

## Figures and Tables

**Figure 1 molecules-28-01623-f001:**
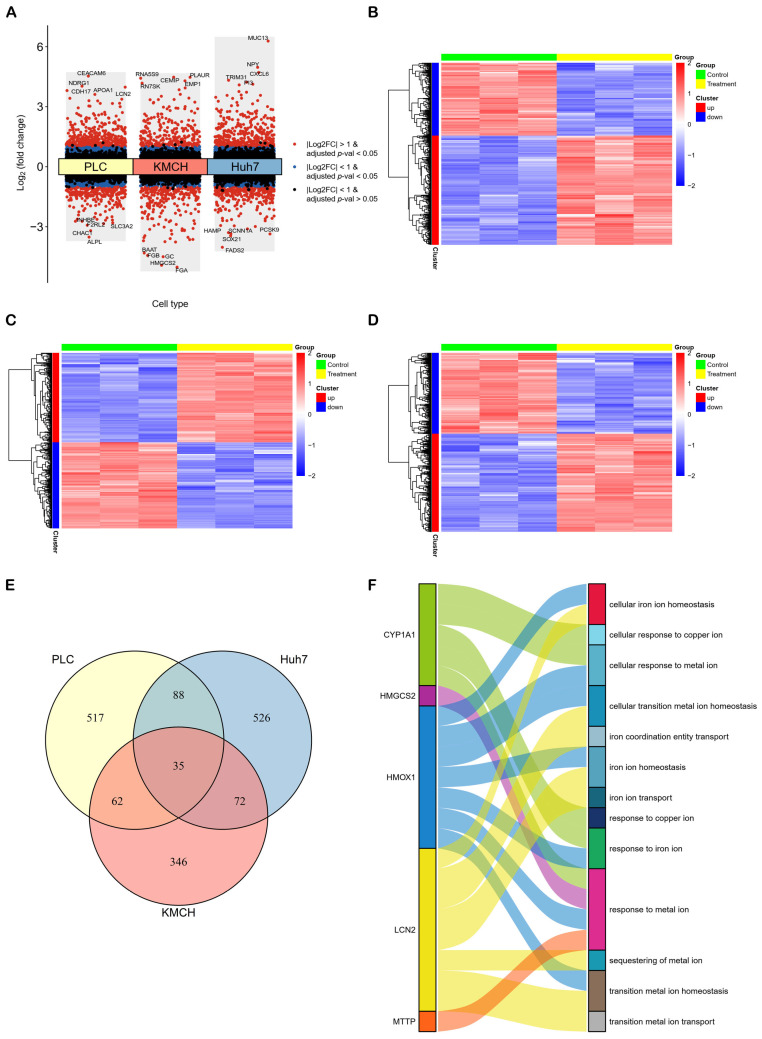
Differential expression analysis and functional annotation. (**A**) A volcano plot and (**B**–**D**) three heatmaps of DEGs showed that significant changes in mRNAs caused by curcumin in PLC, KMCH and Huh7 cell lines, respectively. (**E**) Thirty-five common genes were significantly regulated by curcumin in all three cell lines. (**F**) Among the 35 genes, *CYP1A1*, *HMGCS2*, *HMOX1*, *LCN2* and *MTTP* were enriched in biological processes related to metal ion homeostasis.

**Figure 2 molecules-28-01623-f002:**
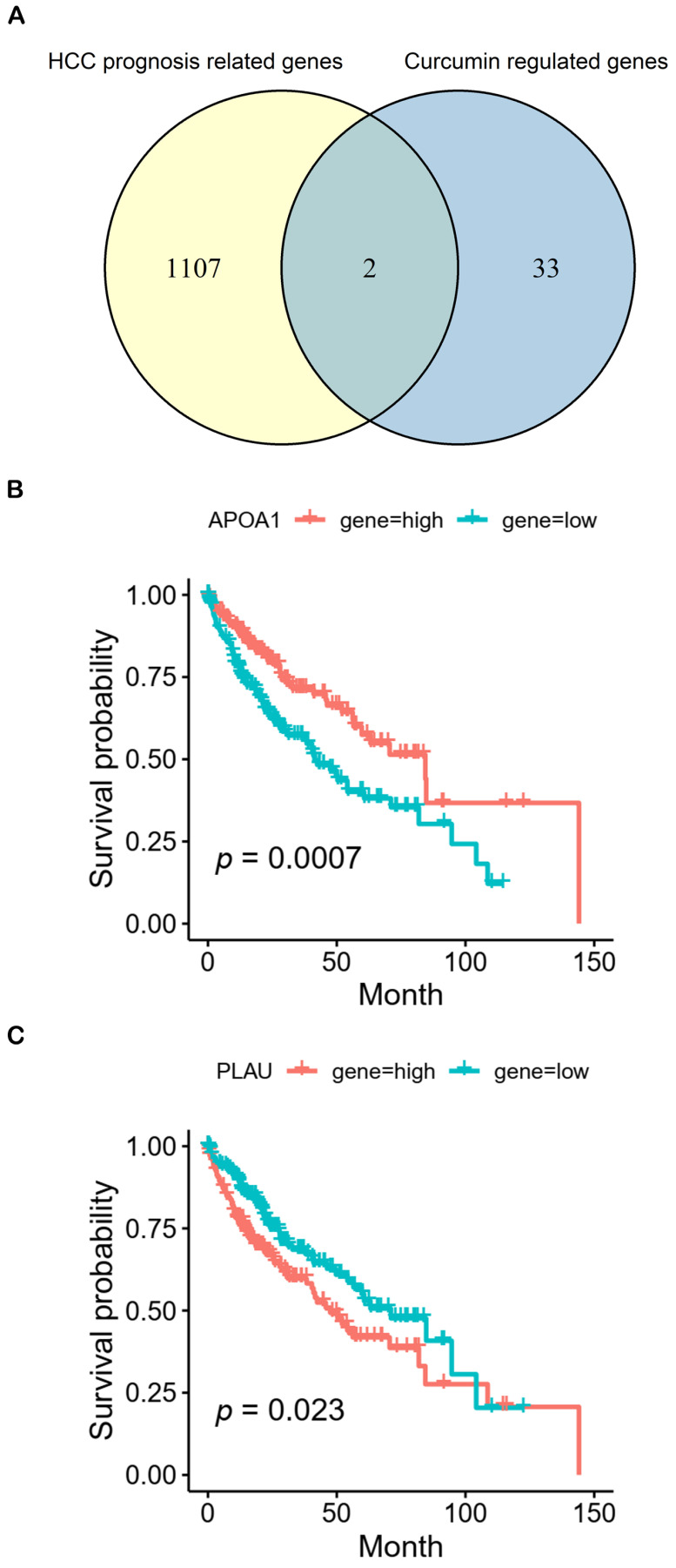
Prognostic analysis. (**A**) There are two common genes (*APOA1* and *PLAU*) between 1109 genes (*p* < 0.05) related to prognosis of HCC (yellow) and 35 DEGs (blue). (**B**,**C**) Kaplan-Meier analysis of overall survival for *APOA1* and *PLAU*, respectively.

**Figure 3 molecules-28-01623-f003:**
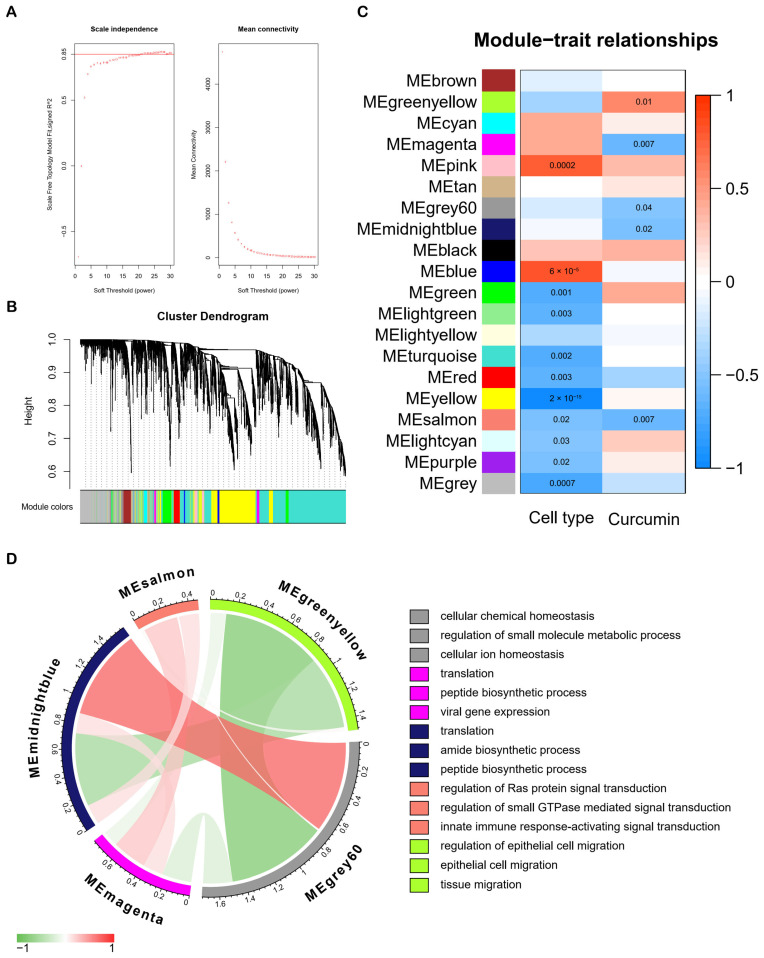
WGCNA analysis and module annotation. (**A**) The soft threshold is set to 21 for WGCNA analysis. (**B**) Twenty modules were constructed through hierarchical clustering. (**C**) The heatmap showed the correlations between modules and traits. (**D**) Correlations among five modules related to curcumin treatment and annotations.

**Figure 4 molecules-28-01623-f004:**
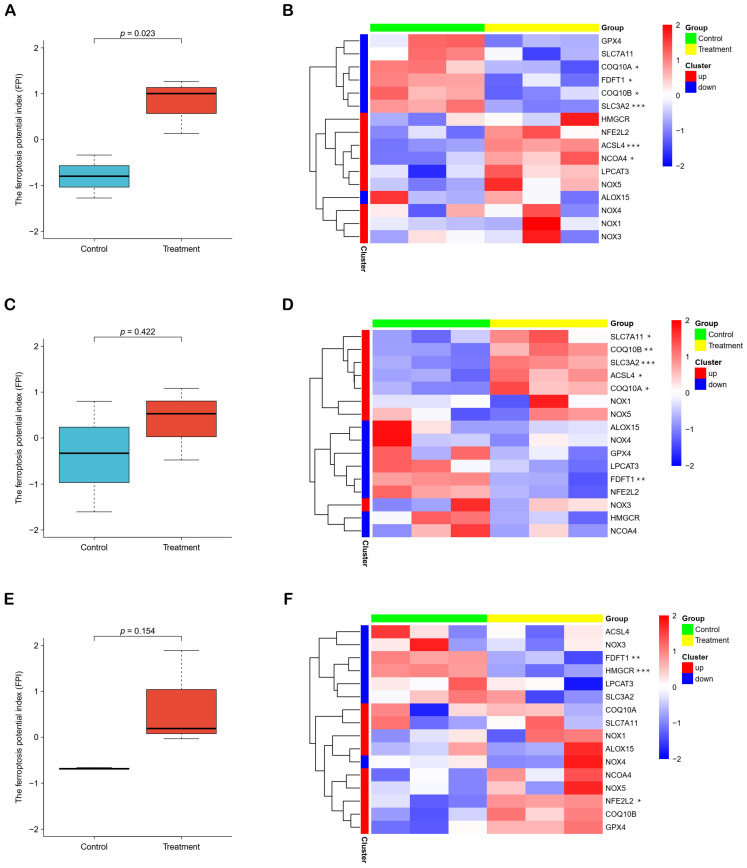
The ferroptosis potential index. (**A**,**C**,**E**) Three boxplots showed the levels of ferroptosis in PLC, KMCH and Huh7 cell lines after curcumin treatment, respectively. (**B**,**D**,**F**) Three heatmaps showed the expression of genes used to calculate FPI. * *p* < 0.05. ** *p* < 0.01. *** *p* < 0.001.

**Figure 5 molecules-28-01623-f005:**
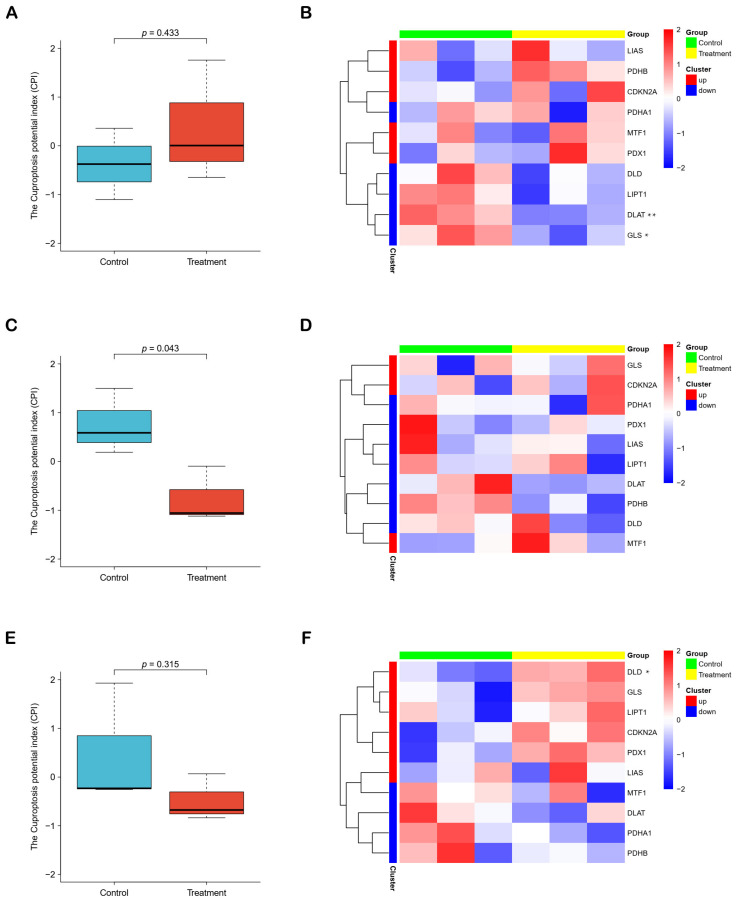
The cuproptosis potential index. (**A**,**C**,**E**) Three boxplots showed the levels of cuproptosis in PLC, KMCH and Huh7 cell lines after curcumin treatment, respectively. (**B**,**D**,**F**) Three heatmaps showed the expression of genes used to calculate CPI. * *p* < 0.05. ** *p* < 0.01.

**Figure 6 molecules-28-01623-f006:**
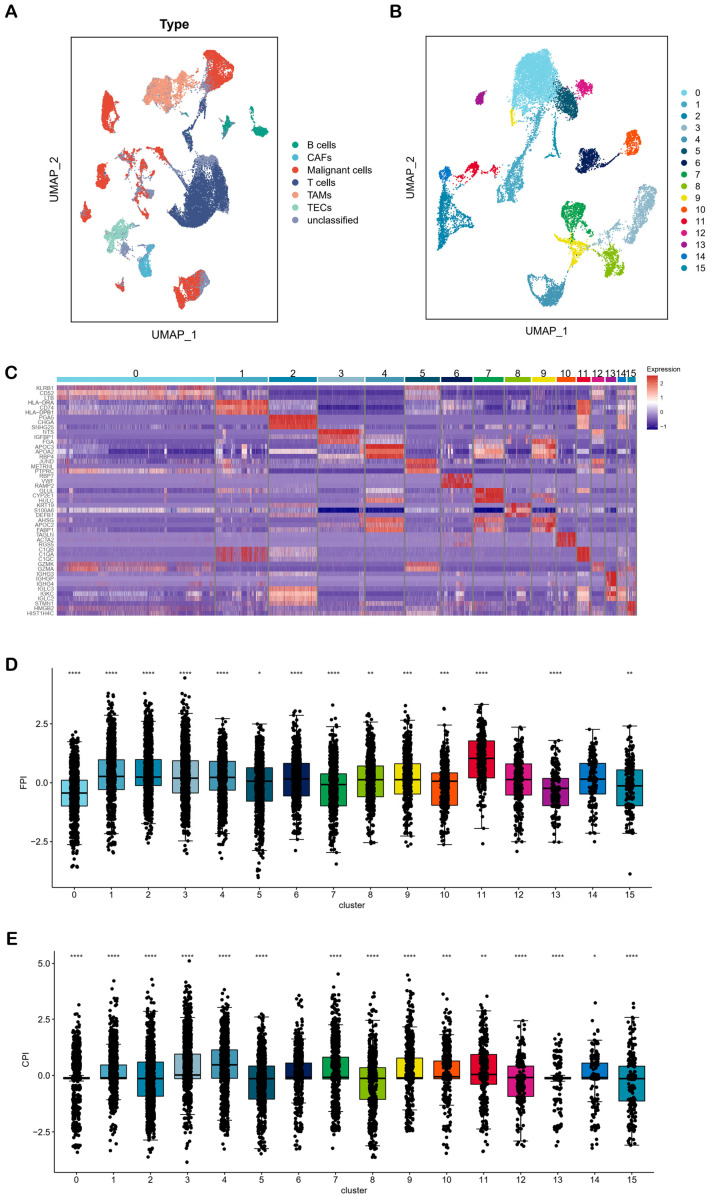
Single cell transcriptome analysis of HCC. (**A**) The UMAP plot of 56,721 single cells from 46 liver tumor samples showed 7 cell types, including B cells, cancer-associated fibroblasts (CAFs), malignant cells, T cells, tumor-associated macrophages (TAMs), tumor-associated endothelial cells (TECs) and unclassified. (**B**) The UMAP plot showed 16 subcluster of malignant cells. (**C**) The heatmap showed the marker genes with significant differences among the 16 subclusters. (**D**,**E**) FPI and CPI were significantly different among the 16 subclusters. * *p* < 0.05. ** *p* < 0.01. *** *p* < 0.001. **** *p* < 0.0001.

## Data Availability

The datasets analyzed during the current study are available in the GEO repository, GSE59713 and GSE151530.

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
