# Peer review of "The Role of Ferroptosis and Cuproptosis in Curcumin against Hepatocellular Carcinoma"

_molecules, 2023, doi:10.3390/molecules28041623_

Round 1

Reviewer 1 Report

The submitted paper describe the results of differential gene expression analysis and weighted gene co-expression network analysis to explore the therapeutic mechanism of curcumin against hepatocellular carcinoma (HCC). The submitted paper would be suitable for publication in Molecules because of the reported changes in gene expression related to metal ion homeostasis induced by curcumin administration. However, there are several points that should be discussed before the publication. Please consider adding a discussion based on the comments below.

1.     Figure 2B showed that high expression of APOA1 and low expression of PLAU were related to the good prognosis of HCC (lines 100-102). However, these genes are not among the genes related to metal ion homeostasis shown in Figure 1F. The prognosis effects of APOA1 and PLAU may be a different mechanism than the anti-cancer effects derived from ferroptosis and cuproptosis.

2.     Among the proteins encoded by genes shown in Figure 1F, only the role of HMOX1 with respect to ferroptosis was discussed (lines 196-206). Therefore, the involvement of proteins other than HMOX1 on ferroptosis should also be discussed. For example, Q. Zhang et al. proposed that MTTP may be a potential protective factor against ferroptosis. (Advanced Science, 2022, 9, 220335, https://doi.org/10.1002/advs.202203357). Also, there have been several reports on the relationship between LCN2 expression and ferroptosis (M. R. Valashedi et al., Life Sciences, 2022, 304, 120704, https://doi.org/10.1016/j.lfs.2022.120704; F. Yao et al., Nature Communication, 2021, 12, 7333, https://doi.org/10.1038/s41467-021-27452-9). Additionally, if possible, please discuss the impact of these proteins on cuproptosis as well as ferroptosis.

3.     According to the description in the discussion section (lines 207-215), curcurmin administration inhibited cuproptosis in KMCH cells. However, inhibition of cuproptosis would lead to suppression of cupper-induced cell death, resulting in a failure of anti-cancer effect of curcurmin.

Reviewer 2 Report

In this study, the authors explored the differential expressed genes between the three types of HCC cells (PLC, KMCH and Huh7) treated with curcumin. Based on ferroptosis potential index (FPI) and the cuproptosis potential index (CPI), the activity of inducing ferroptosis and cuproptosis by curcumin treatment was subsequently analyzed. In addition, the single-cell RNA analysis based on GEO database was also performed, which showed the heterogeneity of HCC cells may be the underlying mechanism by which curcumin plays different roles in inducing ferroptosis and cuproptosis in different cells.

Here are several questions and/or suggestions the authors may consider.

1. My major concern is, it seems the authors only provided the FPI and CPI as the evidence to support their major conclusion that curcumin induced ferroptosis in HCC cells. I believe the authors should at least demonstrated several characterized events ferroptosis, which are distinct from other regulated cell death phenotypes such as apoptosis and necroptosis (such as depletion of glutathione, elevated levels of lipid reactive oxygen species, etc)

2. Similarly, the authors showed that curcumin treatment only significantly decrease the CPI values in KMCH cells. Does this mean that curcumin showed little effects on KMCH viability? The authors should provide the overall killing effects of curcumin on these three cells and discuss other possible mechanisms by which curcumin inhibits the proliferation of HCC cells.

3. Moreover, is there any thresholds for FPI and CPI? The significant increase only shows the difference is statistically different, but does not mean there exists real bioactivity.

Round 2

Reviewer 1 Report

The revised manuscript makes appropriate additions to the discussion according to earlier comments and is more informative about the contribution of ferroptosis and cuproptosis to the therapeutic mechanism of curcumin against hepatocellular carcinoma. Therefore, the revised manuscript would be acceptable for publication in its present form.

Reviewer 2 Report

The authors have addressed all my questions. I would like to recommend its acceptance.